# The Effect of Lower Limb Combined Neuromuscular Electrical Stimulation on Skeletal Muscle Signaling for Glucose Utilization, Myofiber Distribution, and Metabolic Function after Spinal Cord Injury

**DOI:** 10.3390/ijerph20206958

**Published:** 2023-10-21

**Authors:** Amal Alharbi, Jia Li, Erika Womack, Matthew Farrow, Ceren Yarar-Fisher

**Affiliations:** 1Department of Physical Therapy, University of Alabama at Birmingham, Birmingham, AL 35233, USA; amal9@uab.edu; 2Department of Physical Medicine and Rehabilitation, Ohio State University, Columbus, OH 43210, USA; jia.li@osumc.edu (J.L.); matthew.farrow@osumc.edu (M.F.); 3Department of Biochemistry, Molecular Biology, Entomology and Plant Pathology, Mississippi State University, Starkville, MS 39762, USA; ewomack@mscl.msstate.edu; 4Department of Neuroscience, Ohio State University, Columbus, OH 43210, USA

**Keywords:** spinal cord injury, neuromuscular electrical stimulation, metabolic function, myofiber type, muscle glucose uptake

## Abstract

Maintaining healthy myofiber type and metabolic function early after spinal cord injury (SCI) may prevent chronic metabolic disorders. This study compares the effects of a 2–5 week combined (aerobic + resistance) neuromuscular electrical stimulation (Comb-NMES) regimen versus a sham control treatment on muscle protein signaling for glucose uptake, myofiber type distribution, and metabolic function. Twenty participants (31 ± 9 years of age) with an SCI (C4-L1, AIS level A–C) within 14 days of the SCI were randomly assigned to control (*N* = 8) or Comb-NMES (*N* = 12). Sessions were given three times per week. Fasting blood samples and vastus lateralis muscle biopsies were collected 24–48 h before or after the last session. Western blots were performed to quantify proteins, immunohistochemical analyses determined muscle myofiber distribution, and enzymatic assays were performed to measure serum glucose, insulin, and lipids. Our main findings include a decrease in fasting glucose (*p* < 0.05) and LDL-C (*p* < 0.05) levels, an upregulation of CamKII and Hexokinase (*p* < 0.05), and an increase in type I (+9%) and a decrease in type IIx (−36%) myofiber distribution in response to Comb-NMES. Our findings suggest that maintaining healthy myofiber type and metabolic function may be achieved via early utilization of Comb-NMES.

## 1. Introduction

The prevalence of metabolic disorders (e.g., type 2 diabetes, impaired glucose tolerance, and abnormal lipid profiles) is higher in those with spinal cord injury (SCI) vs. age- and sex-matched non-injured individuals [1,2]. Individuals with SCI have an accelerated trajectory of metabolic disorders [3,4,5], with chronic disease states occurring at an earlier age compared with non-injured individuals. These chronic diseases may worsen the substantial declines in functional independence, psychological well-being, and quality of life that occur following an SCI [3].

In healthy humans, up to 70% of whole-body glucose metabolism occurs in skeletal muscle. The regulation of glucose homeostasis is dependent on a finely balanced relationship between (i) muscle sensitivity to insulin and (ii) insulin secretion [4,5]. However, following an SCI, the activation and loading levels of skeletal muscle below the level of the lesion are absent or extremely reduced. Without preventive measures, muscle atrophy rapidly occurs and results in a decline in functional capacity. The cross-sectional area of the lower limb muscles is substantially smaller (25–45%) than in non-injured controls at 6 weeks post-injury [6,7]. In addition to fiber atrophy, muscle protein changes occur within several months after an SCI, which ultimately produces a muscle fiber type that is highly fatigable and contains a high proportion of glycolytic fibers that are characterized by a lower resistance to fatigue and impaired oxidative capacity and mitochondrial function [8,9,10,11]. This fiber transformation results in muscle tissue that is insulin-resistant and metabolically inflexible. The muscles of individuals with SCI have similar histochemical and biochemical properties to those with diabetes. This includes fewer type I fibers and a greater proportion of glycolytic (type IIax and IIx) fibers [12]. The glucose handling capacity is impaired due to lower levels of key enzymes, namely glucose transporter 4 (GLUT 4), hexokinase II, glycogen synthase, and pyruvate dehydrogenase-E1α. This downregulation of GLUT4 expression has been associated with lower whole-body glucose tolerance and insulin sensitivity in individuals with chronic SCI [12].

These observations support the idea that rather than attempting to reverse changes in the sub-acute or chronic stage of SCI, early interventions are likely to more effective in the prevention of deleterious muscle adaptations. One potential and promising strategy to improve skeletal muscle metabolism is neuromuscular electrical stimulation (NMES). NMES-resistance training has been shown to increase the size and distribution (type IIa) of myofibers [13]. However, the distribution of type I fibers did not increase after eight weeks. This highlights the need for an NMES program that produces key adaptations akin to resistance and aerobic exercise to maintain a muscle phenotype that is fatigue-resistant, insulin-sensitive, and oxidative.

To identify effective strategies for improving metabolic profiles among individuals with SCI, we aimed to test the impact of an early intervention (14 days after SCI) of a combined (aerobic + resistance) NMES (Comb-NMES) intervention on muscle growth and type distribution, signaling processes involved in muscle glucose uptake, and systemic metabolic responses. We tested the following hypotheses: (1) Comb-NMES will effectively upregulate signaling proteins that promote GLUT-4 translocation; and (2) Comb-NMES will maintain greater muscle size and a healthy muscle fiber phenotype (similar distribution of type I, IIa, and IIx fibers); and (3) Comb-NMES will produce beneficial changes in serum glucose, insulin, and lipid levels.

## 2. Materials and Methods

### 2.1. Participants and Study Design

In a randomized, controlled study design, participants with acute SCI were randomly assigned into intervention or sham-control groups in a 1:1 ratio. 20 individuals (3 F and 17 M; 31 ± 9 years of age; 12 Comb-NMES and 8 controls) with SCI (injury levels: C4-L1 American Spinal Injury Association Impairment Scale (AIS) A–C), within 14 days of the injury, medically stable at the time of testing, with no history of metabolic syndrome and/or type 1 or type 2 diabetes, completed the study (participant characteristics displayed in Table 1). The only medication that was self-reported in either group was anti-spasticity medication (Baclofen). Outcome measures were taken at two time points: (1) during inpatient rehabilitation (IPR) within the first 14 days of injury and within week 1 of initial rehab; and (2) pre-discharge at week 4 or 5. The Comb-NMES group received a Comb-NMES exercise, and the control group received a sham exercise in addition to their IPR standard care. Standard care included respiration therapy, bowel and bladder management, tone and spasticity management, bed mobility, transfers, wheelchair mobility skills, and skills for performing activities of daily living.

### 2.2. Interventions

#### 2.2.1. Comb-NMES Intervention

Comb-NMES included NMES-resistance exercise and NMES-aerobic exercise three days per week (M, W, and F), separated by a rest day. NMES resistance required concentric and eccentric quadricep contractions while seated. Each session had four sets of ten repetitions that were elicited using 450 µs biphasic pulses at 50 Hz. Surface NMES was used to carry out the protocol. Tetanic muscular contractions that evoke complete knee extension were induced by raising the current from zero to the goal level (50–200 mA) in 3 to 5 s. Following the production of the knee extension, the current was quickly reduced (an additional 3–5 s) to allow for progressive relaxation into a flexed position. There was normally a 5 min break between sets because both legs were worked in an alternating fashion [13,14,15]. The weights were progressively increased by 2 lb. on a weekly basis after achieving 40 repetitions of full knee extension in a session. If participants failed to obtain full knee extension for 4 sets of 10 repetitions, the same load was maintained until the desired repetitions were achieved. During the first week, ankle weights were not used to ensure full knee extension against gravity and to prevent skeletal muscle damage [16,17]. The current amplitude of electrical stimulation was monitored for each repetition to ensure that the increase in weight lifted was a result of muscle adaptations and not increased electrical stimulation current amplitudes (Figure 1A). Upon completion of each NMES-RE session, participants were given a short break (10–15 min) for recovery before starting aerobic training. NMES aerobic exercise involved twitch electrical stimulation (pulse duration/interval = 200/50 µs) applied to the quadriceps muscle via surface NMES. The current amplitude was set to 100–200 mA [18] and kept consistent for each patient. The training started with 10 min of twitch stimulation at 2 Hz. After the first week, the duration of the session was progressively increased to 60 min at 10 Hz (Figure 1B).

#### 2.2.2. Sham Intervention

This group received standard care plus passive dynamic exercise of the lower legs (sham treatment for NMES-RE) and transcutaneous electrical nerve stimulation (TENS, sham treatment for NMES-aerobic exercise) during IPR. For the passive dynamic lower leg exercise, a physical therapist passively moved the participant’s lower leg to a full knee extension and flexion for a similar duration used for NMES-RE (3–5 s to attain knee extension and flexion). Both legs were trained in an alternating order with 5 min of rest between sets. Upon completion of leg exercises, participants were given a short break (10–15 min) for recovery before starting the TENS intervention. For TENS, we applied 1/4 of the current needed to aerobically train the muscle. For example, 100–200 mA is typically needed to adequately stimulate the paralyzed muscle with long-standing SCI; thus, 25–50 mA with a pulse duration/interval of 200/50 µs was used as the control stimulus [8]. This protocol allowed some current to flow; however, it did not result in visible muscle contractions. All participants in both groups were either trained in their wheelchairs or beds.

### 2.3. Clinical and Laboratory Procedures

#### 2.3.1. Muscle Biopsy

Our lab’s established percutaneous needle biopsy technique was used to obtain samples from the vastus lateralis [18,19]. Biopsies were carried out utilizing a 5-mm Bergstrom-type biopsy needle while under local anesthesia (1% lidocaine). For immunohistochemistry, 50–70 mg of muscle was mounted cross-sectionally and frozen in liquid nitrogen-cooled isopentane. For upcoming biochemical tests, the remaining tissue was snap-frozen in 30 mg pieces. The list of primary and secondary antibodies used in the fluorescent immunohistochemistry and immunoblotting appeared in Table 2.

#### 2.3.2. Myofiber-Type Distribution

The relative distributions of myofiber types I, IIa, and IIax/IIx were determined immunohistochemically using our laboratory’s well-established protocol [20]. Briefly, at −20 °C, the frozen muscle mounts were sliced into 6 µm cross-sections using a Leica CM1860 cryostat chamber (Deer Park, IL, USA). Sections were cut in triplicates and mounted on three-well slides (Electron Microscopy Sciences, Hatfield, PA, USA) before IHC staining. Cryostat sections were air-dried at room temperature for 30 min before being kept in a humidified chamber throughout the staining protocol. NCL-MHCs (anti-MHC I) and NCL-MHCf (anti-MHC II) primary antibodies were obtained from Leica Biosystems (Deerfield, IL), and anti-laminin antibodies were obtained from Sigma-Aldrich (St. Louis, MO, USA). Thermo Fisher Scientific (Norcross, GA, USA) provided secondary antibodies, ALEXA Fluor 594 (red fluorescent dye) and 488 (bright green fluorescent dye), and 3% neutral-buffered formalin. According to the manufacturer’s instructions, primary and secondary antibodies were diluted in 1% goat serum. Tissue sections were fixed for 30 min in 3% neutral-buffered formalin. To decrease non-specific background staining, sections were blocked with 5% goat serum prior to the application of primary and secondary antibodies and washed 3× for 5 min with ice-cold 1x phosphate-buffered saline (PBS; pH = 7.4). ALEXA Fluor 594 (MHC type I, slow) and 488 (MHC type II, rapid) were used to stain the fiber types. The anti-laminin antibody was co-stained with ALEXA Fluor 488 to enable observation of fiber boundaries and myofiber size. Sections mounted with Vectashield mounting fluid containing DAPI (Vector Laboratories, Burlingame, CA, USA) created a blue fluorescence that allowed nuclei to be identified. Slides and coverslips were bonded together with nail varnish and kept at −20 °C until microscopy. In the present study, we pooled hybrid IIax fibers with type IIx fibers due to the high percentage of IIax myofibers found in the samples. For the Comb-NMES group, myofiber type distribution was determined for 7641 (min: 256, max: 1328) VL myofibers pre-training and for 6978 (min: 379, max: 1355) VL myofibers post-training. Myofiber-type distribution was determined for 4550 (min: 603, max: 1056) VL myofibers pre-training and 5164 (min: 309, max: 1618) VL myofibers post-training for the control group.

#### 2.3.3. Immunoblotting

Mixed muscle protein lysate was prepared using an established method in our laboratory [21]. We homogenized muscle samples (30 mg) after a 15 min preincubation in ice-cold lysis buffer with protease and phosphatase inhibitors and centrifuged them for 10 min at a temperature of 4 °C at 15,000× *g*. The supernatant was stored at −80 °C until assayed for protein quantity using the bicinchoninic acid technique with bovine serum albumin (BSA) as a standard.

Skeletal muscle total protein (20 µg) lysate was resolved on 4–12% SDS-PAGE gels using our laboratory protocol as described previously [21,22]. All gels contained samples from both the intervention and control groups loaded in series. The transfer of the proteins from the gels to the nitrocellulose membranes was performed using the Thermo Scientific Pierce Power Blotter preprogrammed transfer methods. Total lane normalization was achieved by obtaining imaging of the Ponceau S-stained membranes using the Bio-Rad ChemiDoc imaging system immediately after transfer to verify the total protein from each lane. The content and phosphorylation of proteins associated with skeletal muscle GLUT-4 translocation were assessed. The membranes were blocked for 60 min at room temperature in 5% non-fat milk in TBST (0.1% Tween 20 and 150 mM NaCl in 10 mM Tris-Base, pH 7.4) and probed overnight at 4 °C with the targeted primary antibody purchased from Thermo Scientific (Rockford, IL, USA) and Cell Signaling Technologies (Danvers, MA, USA) at a concentration of 1:1000 dilution in 5% BSA with TBST (*v*/*v*) against the following: GLUT-4 transporter total AMPK-α, phospho (Thr172)-AMPK-α, total CaMKII, total Akt, phospho (Thr286)-CaMKII, phospho (Thr642)-AS160, phospho (Ser473)-Akt, total AS160, Hexokinase II, Glycogen Synthase (GS), phospho (Ser641)-GS, IRS-1, and phospho (Tyr895)-IRS-1. Horseradish peroxidase-conjugated secondary antibodies were used at a concentration of 1:2500, followed by chemiluminescent detection (SuperSignal West Femto Chemiluminescent Substrate, Thermo Scientific, Rockford, IL, USA) in a BioRad ChemiDoc imaging system with band densitometry performed using BioRad Image Lab software (version 6.1). The density of the band was measured against the total lane background to obtain the normalized band volume.

#### 2.3.4. Determination of Blood Glucose, Insulin, and Lipids

All analyses took place at the UAB Nutrition Obesity Research Center-Metabolism Core. Serum glucose, total cholesterol, LDL, HDL, and triglycerides were assessed using an automated analyzer (Sirrus analyzer; Stanbio Laboratory, Boerne, TX, USA). LDL concentrations were calculated using the Friedwald method [23]. Serum insulin was measured using immunofluorescence with an AIA-600 II analyzer (TOSOH Bioscience, South San Francisco, CA, USA).

### 2.4. Statistical Analysis

Descriptive statistics for subjects’ characteristics were calculated with a mean ± standard deviation for normally distributed continuous variables. Median and interquartile range (IQR) are reported for non-normally distributed variables (i.e., duration). Normality was evaluated using the Shapiro-Wilk test. Unpaired *t*-tests and Mann-Whiteney U tests were used to compare age and duration between groups, respectively. Fisher’s exact test was used to compare baseline group characteristics for continuous and categorical variables, respectively. A linear mixed model analysis was constructed to test the effect of treatment, time, and their interaction on the participant as a random effect using the SAS 9.4 proc mixed procedure (SAS Institute, Inc., Cary, NC, USA). Outcomes of interest for the linear mixed-effects model include indices of glucose homeostasis (i.e., fasting glucose, insulin), lipid profile (i.e., total cholesterol, triacylglycerol, HDL-cholesterol, LDL-cholesterol), skeletal muscle proteins involved in GLUT4 translocation, and muscle fiber type distribution. Pairwise post hoc comparisons were performed using the Tukey–Kramer multiple comparisons method to estimate within-group changes within the linear mixed-effects model. Statistical model assumptions (e.g., homogeneity of variance, normal distribution of residuals) were confirmed before data analysis using the respective diagnostic plots. Statistical tests were two-sided, with *p* < 0.05 considered statistically significant. Statistical analyses were performed using SAS, version 9.4 (SAS Institute, Inc.). Data are presented as means ± standard errors from the linear mixed-effects model unless otherwise stated. In the current report, due to the small sample size, we did not include covariates to avoid overfitting the statistical model. Furthermore, in our preliminary analysis, including age, level of injury, completeness, and duration of stay did not change our main observations. We could not obtain sufficient muscle samples from 5 individuals and blood samples from 6; therefore, final analyses were done on 15 people (Comb-NMES: 8 and control: 7) for muscle intracellular signaling and fiber-type and 14 (Comb-NMES: 8 and control: 6) for glucose, insulin, and lipid outcomes.

## 3. Results

### 3.1. Participant Characteristics

Participants’ demographic data are shown in Table 1. Overall, there were no significant differences between groups in terms of the distribution of factors such as level of injury, completeness, injury type, gender, and race. The median duration of intervention for the control and Comb-NMES groups was 18 and 21 days, respectively (*p* = 0.38). The mean age of participants in the Comb-NMES group was higher than that of the control group (age difference: 9.8 years, *p* = 0.01). The average intensity delivered in the Dudley and Twitch protocols in the Comb-NMES group was 123.8 mA, ±45.2, and 118.2 mA, ±44.2, respectively. In addition, the average repetition was given in the Dudley protocol as 9 ± 2 reps.

### 3.2. Glucose Homeostasis (N = 14)

A significant time effect in fasting glucose concentration was observed, where fasting glucose concentrations decreased over time regardless of the treatment group (*p* < 0.05 for time effect). However, there was no interaction (group × time) effect (*p* = 0.15). Post hoc analysis showed that the changes were significant (−19.9 mg/dL, *p* = 0.01) in Comb-NMES but not significant in the control group (−7.3 mg/dL, *p* > 0.05; Figure 2A). Similarly, a significant time effect was observed for fasting insulin, where insulin decreased over time regardless of the treatment group (Figure 2B; *p* < 0.05 for time effect). Changes in fasting insulin within each group were not significant.

### 3.3. Lipid Profile (N = 14)

There were no changes in total cholesterol, triacylglycerol, or HDL (Figure 3A–C). A significant interaction (group × time) effect was found for LDL cholesterol (*p* < 0.05). The interaction effect was driven by a decrease in LDL cholesterol for the Comb-NMES group (pre: 109.8 mg/dL vs. post: 89.3 mg/dL) and an increase in the control group (pre: 78.5 mg/dL vs. post: 92 mg/dL; Figure 3D), though only the changes in the Comb-NMES were statistically significant (*p* = 0.01).

### 3.4. Skeletal Muscle Intracellular Signaling (N = 15)

All of the putative proteins and their relevant phosphorylation involved in the GLUT4 translocation pathway did not change differently over time between groups except for the phosphorylation of GS_Ser641. Phosphorylation of GS_Ser641 decreased in the Comb-NMES group and increased in the control group (interaction: *p* < 0.05; changes within each group were not statistically significant, Figure 4B). CamKII, AKT, pAKT_Ser473, and Hexokinase II increased over time regardless of treatment group (time: *p* < 0.05, Figure 4C,E–G). There was a tendency towards a time effect for Glut4 (time: *p* = 0.07) and IRS-1 (time: *p* = 0.08; Figure 4H–I), where Glut4 and IRS-1 increased over time. Despite the lack of interaction effect, the increase for some of these proteins was higher in the Comb-NMES group. For example, there was a 170% increase in Hexokinase II in the Comb-NMES (*p* < 0.05) group after the intervention compared with a 22% increase in the control group. There was a 67%, 49%, 133%, 63%, and 32% increase in CamKII (*p* < 0.05), Glut4, IRS-1, AS160, and GS in the Comb-NMES group after the intervention, compared with a 16%, 36%, 14%, 3%, and 17% increase in the control group, respectively. Representative immunoblots of the skeletal muscle intracellular signaling proteins are presented in Figure 5. Additional immunoblots can be found in the Appendix A.

### 3.5. Myofiber-Type Distribution (N = 15)

There was no effect of intervention, time, or their interaction on myofiber type distribution (Figure 6). However, MHCI increased by 9% in the Comb-NMES group compared with a 12% reduction in the control group. In addition, MHC IIx decreased by 36% in the Comb-NMES group compared with a 67% increase in the control group.

## 4. Discussion

People with SCI are more susceptible to type II diabetes, obesity, and cardiovascular disease compared with age- and sex-matched non-injured individuals [1,2]. Maintaining sufficient muscle mass and metabolic function is likely to be a potent strategy to prevent a decline in cardiometabolic health in individuals with SCI. Implementing an early intervention to prevent these deleterious adaptations may work more effectively than attempting to reverse changes several months or years after SCI. Therefore, our objectives were to test the impact of early intervention (14 days after SCI) of a Comb-NMES intervention on muscle growth and type distribution, signaling processes that regulate muscle glucose uptake, and systemic metabolic responses. This is the first study to use a novel Comb-NMES approach coupling electrically induced resistance and aerobic exercise on the knee extensor muscle group (quadriceps) to maintain or even improve muscle and systemic metabolic function. The primary and novel findings are that Comb-NMES intervention resulted in more notable increases in total protein levels of IRS-1 (+133% vs. +14%), CamKII (+67% vs. +16%), Hexokinase II (+170% vs. +22%), and GS (+37% vs. −24%). We also showed that Comb-NMES intervention successfully increased slow-oxidative type I fibers (+9%) and decreased fast-glycolytic type IIx fibers (−36%). On the other hand, slow-oxidative type I fibers decreased (−12%), and fast-glycolytic type IIx fibers increased (+67%) in response to the control intervention. In addition, there was a significant and more considerable decrease in fasting blood glucose levels in the Comb-NMES (*p* < 0.05) compared with the control group (−19.9 mg/dL vs. −7.3 mg/dL). Although no changes were observed in total cholesterol, triglycerides, or HDL, LDL cholesterol significantly decreased in the Comb-NMES group (−20.5 mg/dL, *p* < 0.05) compared with an increase in the control group (+13.5 mg/dL). These results attest to the robust systemic and muscle responses to Comb-NMES.

### 4.1. Glucose, Insulin, and Lipid Profile

Reduced fasting glucose (−19.9 mg/dL) and insulin levels (−11.29 mg/dL) in response to 2–5 weeks of Comb-NMES are promising for promoting glucose homeostasis in acute SCI. This finding agrees with our previous work in individuals with long-standing SCI [24] that demonstrated improvements in fasting glucose levels following an 8-week NMES-RT. Other studies [2,24] showed no effect of NMES-RT on fasting glucose and insulin levels in response to 12 to 16 weeks of NMES-RT. Given that previous interventions were significantly longer than the present intervention (12–16 weeks vs. 2–5 weeks) and used only RT as an intervention, repetitively stressing the muscle with low-frequency electrical stimulation (aerobic training) might be needed for marked improvements in glucose homeostasis. The type of skeletal muscle fibers affects insulin sensitivity throughout the body. A positive correlation exists between the proportion of type I fibers in muscle and whole-body insulin sensitivity in humans. When [25] type I fibers were predominant, human muscle strips showed greater insulin-stimulated glucose transport [26]. Hence, type I fibers are likely to play a greater role in maintaining glucose homeostasis in response to insulin than type II fibers. Furthermore, in humans, GLUT4 protein levels are highest in type I fibers (vs. IIa and IIx) [27,28]. These findings suggest insulin’s effect on glucose transport is highest in type I fibers. Our study found that type I fibers and GLUT-4 protein levels increased in response to Comb-NMES training. Therefore, the reduction in fasting glucose in the Comb-NMES group may be due to an increase in type I fibers.

LDL-cholesterol is a well-established cardiovascular disease risk factor for cardiovascular disease, with evidence from large-scale epidemiological and Mendelian randomization studies [29,30]. LDL cholesterol levels significantly changed in both groups. It decreased in the Comb-NMES group (−20.5 mg/dL) and increased in the control group (+13 mg/dL, Figure 3D). Our finding of reduced LDL cholesterol agrees with a previous study that demonstrated a reduction in LDL cholesterol (−7.2 mg/dL) in response to 12-week, twice-per-week NMES-RT [31]. Other studies have not shown marked changes in lipid profiles when administered NMES-RT combined with testosterone treatment [32] or home-based NMES endurance training for 16 weeks [33]. In addition, no change in lipid profile was reported with an 8-week, three times per week upper extremity aerobic exercise combined with NMES-RT in individuals with long-standing SCI [24]. Therefore, early intervention with Comb-NMES might be needed to reduce LDL cholesterol significantly. Although participants in this study did not present with cardiovascular disease risk factors, a 1-mmol/L (»18 mg/dL) reduction in LDL-C with statin therapy was associated with a 23% reduction in major vascular events in >300,000 participants with major vascular events [29].

### 4.2. Skeletal Muscle Intracellular Signaling

The regulation of muscle glucose utilization consists of three serial steps: the delivery of glucose from the blood to the muscle membrane, the transport across the muscle membrane, and the phosphorylation of glucose-6-phosphate. Upon stimulation by insulin [34] or exercise [35], GLUT4 translocates to the sarcolemma, making it highly permeable to glucose. GLUT4 translocation lowers the barrier to glucose transport and, in effect, shifts the responsibility to glucose delivery, phosphorylation, and flux through muscle metabolism. We previously showed that GLUT-4 and IRS-1 total protein content was significantly lower in the resting muscles of individuals with long-standing SCI compared with age- and sex-matched non-injured individuals [36], suggesting a barrier in glucose transport across the muscle membrane. The total protein content for GLUT-4, IRS-1, Hexokinase II, CaMKII, and GS total protein content increased in the Comb-NMES group, suggesting that early and chronic exposure to Comb-NMES training improved both the insulin- and contraction-stimulated translocation of GLUT4 in the paralyzed muscle.

The translocation of GLUT-4 is caused by muscle contractions caused by the activation of AMPK [37,38] and CaMK-mediated signaling pathways [39]. Additionally, insulin and contraction-induced GLUT-4 translation in skeletal muscle is mediated by AS160 [40]. However, we found no alteration in the overall protein levels of AMPK in the Comb-NMES group. This could be attributed to changes in the distribution of muscle fiber types. Previous studies have reported differences in AMPK activation between different fiber types at rest and after a single exercise session [41]. Resting AMPK phosphorylation seems to be the most significant in type IIa fibers, while a single exercise session leads to increased AMPK phosphorylation in all fibers, particularly type IIx fibers. Therefore, the absence of changes in AMPK observed in our study may be due to an increase in type I fibers and a decrease in type IIx fibers in response to Comb-NMES. Previous research suggests that the increase in cytosolic Ca^2+^ during muscle contractions also contributes to pathways involved in increased glucose uptake by the muscles [39,42]. Ca^2+^ stimulates glucose transport through the activation of CaMK, specifically CaMKII, which is the primary isoform of CaMK found in skeletal muscle [43]. Although there is evidence that CaMKII regulates glucose uptake through AMPK-dependent mechanisms [44], a growing body of evidence indicates that CaMKII-dependent signaling regulates skeletal muscle glucose uptake independent of AMPK, Akt, and AS160 phosphorylation [45,46]. In our previous work, we showed higher activation of CaMKII in response to an acute bout of NMES-RT in individuals with SCI compared with non-injured individuals, which suggests that NMES-RT was sufficient to activate CaMK-dependent pathways [36]. Our recent findings showed an increase in the total levels of CaMKII in both groups, but the increase was much higher in the Comb-NMES group (*p* < 0.05, 67% vs. 16%), suggesting that GLUT-4 translocation is improved via the upregulation of CamKII.

The phosphorylation of glucose by hexokinase is required for glucose uptake in skeletal muscle [47]. The phosphorylated glucose is unable to leave the cell and is the substrate for metabolism. Studies have shown that the robust translocation of GLUT4 to the cell surface in response to exercise or insulin increases permeability to glucose, increasing the contribution of hexokinase II to control muscle glucose uptake [48,49]. While membrane transport via GLUT 4 is considered a primary site for insulin resistance, glucose phosphorylation is also a significant component of muscle insulin resistance. Our findings of a 170% increase in Hexokinase II (*p* < 0.05), a 49% increase in GLUT-4, and an upregulation of IRS-1 and CaMK-II suggest improved glucose transport, uptake, and phosphorylation in the paralyzed muscle in response to Comb-NMES training.

### 4.3. Myofiber Type Distribution

Paralyzed muscle shows a dramatic shift toward increasing IIx fibers soon after injury [50]. Electrical stimulation training increases the size [8], strength [51], and fatigue resistance [52] of paralyzed muscles, as well as inducing a type IIx to IIa fiber type conversion [41]. These adaptations plateau after around 3–6 months of training [41,51]. There is a significant gap in the literature for studies investigating adaptations in myofiber type in response to electrical stimulation after SCI. The transformation from type II to type I fibers has seldom been reported [53], even after 1 year of training [54]. However, an increase in type I mRNA has been observed after 4 weeks of intense, daily, low-frequency stimulation. One study showed that 16 weeks of isometric electrical stimulation training for 60 min, 5 times per week during the first 5 months after injury, was not sufficient to maintain type I fibers in 6 individuals with complete (AIS A) paraplegia (T5–T11) [55]. Participants showed more minor alterations in type I fibers (49% to 40%). Previously, we assessed the effects of 8 weeks of NMES-RE on VL muscle fiber type distribution in individuals with long-standing SCI. Despite the increase in type IIa fiber size, we demonstrated that type I fiber distribution was not changed following NMES-RE. We concluded that low-intensity, chronic aerobic NMES training might be needed to increase type I fiber distribution. As shown by time-series studies and the biochemistry of single fibers [56,57,58,59], when chronic, low-frequency stimulation is applied to a predominantly fast muscle, the fast muscle first changes its metabolic properties, followed by its contractile properties, until it completely transforms into a slow muscle. The present study supports these findings and shows that up to 30 min of 2–7 Hz twitch electrical stimulation per day, three times per week, included a fiber type shift from IIx to type I fibers. Therefore, early prevention via low-frequency chronic stimulation may be the key to maintaining or increasing type I fatigue-resistant and oxidative fibers after SCI. Our low-intensity aerobic NMES protocol was adapted from Ryan et al. [17] and Petrie et al. [60]. Their study showed increased mitochondrial muscle capacity and oxidative phosphorylation following a 24-week, 7 Hz, twitch electrical stimulation (pulse duration/interval = 200/50 µs) of the quadriceps muscle group. Although they did not determine the changes in myofiber type, the 3-fold increase in mitochondrial capacity was substantially more significant than that reported with traditional functional electrical stimulation (FES) exercise programs. This significant change may be due to a marked increase in type I fibers [61,62,63], as electrical activity and muscle properties are intimately interrelated.

Our study has several limitations. Firstly, the small sample size (*N* = 20) may have limited our power to detect significant changes in some of the study outcome measures and further delineate the impact of covariates, such as level of injury and duration of intervention. Secondly, we did not record participants’ habitual dietary intake, and therefore we cannot determine if our interventions elicited any changes in participants’ diet behaviors and if this had an effect on metabolic response. Finally, we did not directly measure glucose transport, uptake, or utilization; therefore, we can only speculate that it is possible that the upregulation of signaling proteins resulted in improved glucose utilization in the paralyzed muscle.

## 5. Conclusions

Our collective data suggest that maintaining healthy myofiber type and metabolic function may be achieved via early utilization of Comb-NMES training in individuals with SCI. Future work should aim to determine the efficacy of home-based Comb-NMES for maintaining or enhancing the metabolic improvements gained during inpatient rehabilitation.

## Figures and Tables

**Figure 1 ijerph-20-06958-f001:**
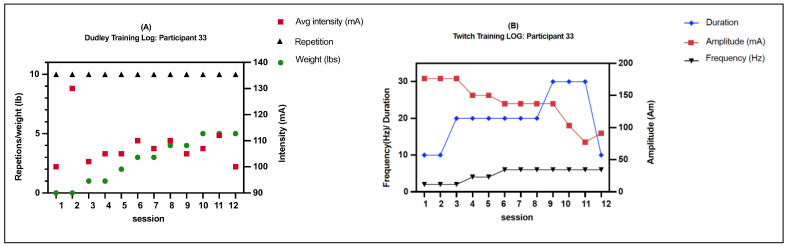
An example of a typical training program for a participant with motor complete SCI. (**A**) Dudley training: Each session included four sets of 10 actions. The participant completed 4 × 10 repetitions in the first two sessions with no added weight. The weights were progressively increased by 1 lb. after achieving 40 repetitions of full knee extension in a session. The participant was able to lift 5 lbs. after 12 sessions. (**B**) Twitch training: The training started with 10 min of twitch stimulation at 2 Hz. After the first week, the duration of the session was progressively increased to 30 min at 7 Hz.

**Figure 2 ijerph-20-06958-f002:**
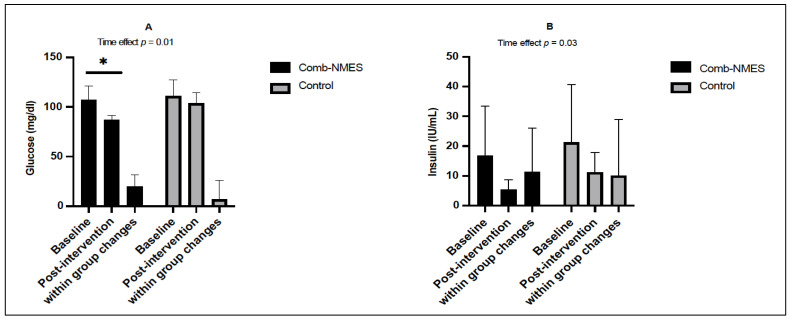
Changes in glucose level (mg/dL) (**A**) and insulin concentrations (IU/mL) (**B**) in response to the interventions in the Comb-NMES and control groups at fasting. Changes within a group refer to the differences between pre- and post-training among individuals in the same group. * Statistically significant changes within the Comb-NMES group. Data are means ± SD.

**Figure 3 ijerph-20-06958-f003:**
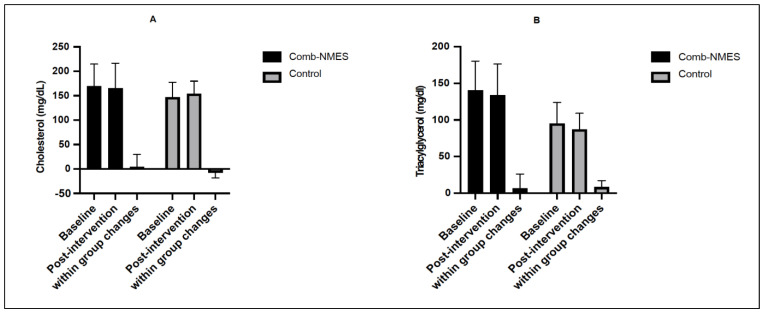
Lipid and lipoprotein levels in Comb-NMES and control at Baseline and Post-intervention. (**A**) cholesterol, (**B**) triacylglycerol, (**C**) high-density lipoprotein HDL, and (**D**) low-density lipoprotein LDL. Changes within a group refer to the differences between pre−and post−training among individuals in the same group. * Statistically significant changes within the Comb-NMES group. Data are means ± SD.

**Figure 4 ijerph-20-06958-f004:**
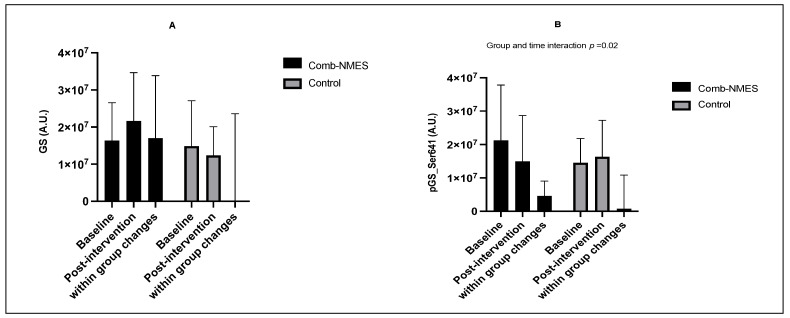
Normalized total (**A**,**C**,**E**,**G**,**H**,**I**,**K**,**M**) and phosphorylated (**B**,**D**,**F**,**J**,**L**,**N**) protein levels for contraction−induced glucose transporter 4 (GLUT−4) translocation signaling pathway in response to Comb-NMES or control. Changes within a group refer to the differences between pre−and posttraining among individuals in the same group. # Statistically significant changes within the group. Data are means ± SD.; Glycogen synthase, GS; Ca + 2/calmodulin-dependent protein kinase, (CaMK) II; protein kinase B, Akt; Insulin receptor substrate 1, IRS-1; AMP-activated protein kinase, AMPK; Akt substrate of kDa, Rab GTPase-activating protein), AS160.

**Figure 5 ijerph-20-06958-f005:**
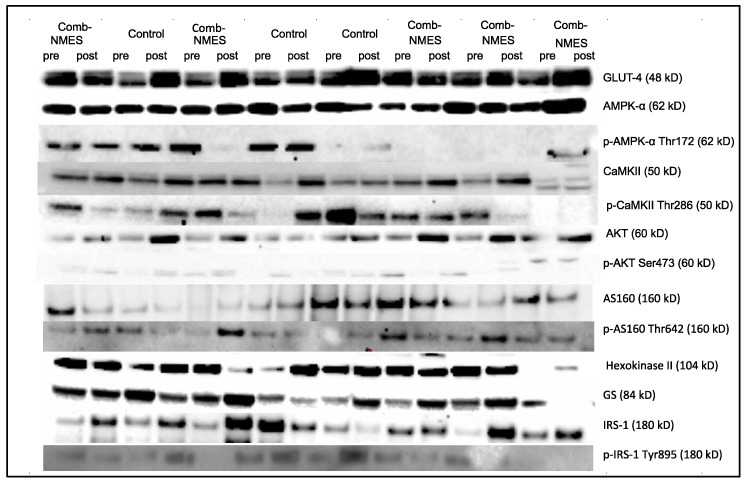
Representative immunoblots for 13 studied proteins in vastus lateralis (VL) muscle samples in the Comb-NMES (*N* = 5) versus control (*N* = 3) group. The samples were loaded in the order shown.

**Figure 6 ijerph-20-06958-f006:**
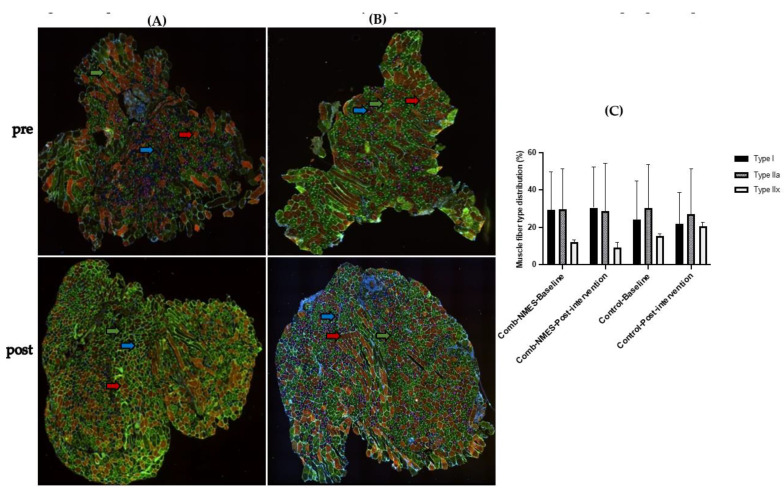
Representative fluorescent immunohistochemistry images from vastus lateralis (VL) muscle samples pre- and post-intervention in (**A**) control and (**B**) Comb-NMES groups. The red arrow indicates type I, the green arrow indicates type IIa, and the blue arrow indicates type IIx. (**C**) Muscle fiber type distribution in response to Comb-NMES and control intervention. Data are means ± SD.

**Table 1 ijerph-20-06958-t001:** Baseline participant characteristics.

Participants	Age	Gender	Race	AIS	LOI	Completeness	Injury Type	Days in Intervention
Control								
SCI 3	29	F	BLACK	A	T2	complete	GSW	10
SCI 4	20	M	BLACK	A	T9	complete	GSW	11
SCI 10	31	M	BLACK	B	C7	incomplete	MVC	20
SCI 16		M	BLACK	A	L1	complete	MVC	24
SCI 19	21	F	BLACK	A	T9	complete	GSW	16
SCI 20	20	M	WHITE	A	T11	complete	FALL	9
SCI 28	28	M	BLACK	A	C6	complete	GSW	36
SCI 31	26	M	WHITE	A	T2	complete	GSW	36
Summary statistics *	25 ± 4.28 *							18 (10.5–30) *
Comb-NMES								
SCI 1	22	M	BLACK	A	C7	complete	GSW	17
SCI 2	25	M	WHITE	B	T6	incomplete	MVC	15
SCI 6	42	M	WHITE	B	C4	incomplete	FALL	20
SCI 11	43	F	WHITE	A	C6	complete	MVC	23
SCI 12	27	M	HISPANIC	A	T7	complete	MVC	21
SCI 17	51	M	BLACK	C	C3	incomplete	MVC	39
SCI 22	41	M	BLACK	A	C4	complete	MVC	21
SCI 24	45	M	WHITE	A	T3	complete	MCC	28
SCI 29	32	M	BLACK	A	T9	complete	GSW	23
SCI 30	34	M	WHITE	A	C7	complete	GSW	21
SCI 32	32	M	WHITE	B	C4	incomplete	MVC	24
SCI 33	24	M	BLACK	A	T3	complete	GSW	17
Summary statistics *	34.8 ± 9.4 *							21 (18.5–23.5) *

SCI, spinal cord injury; AIS, American Spinal Injury Association Impairment Scale; C, cervical SCI; T, thoracic SCI; LOI, level of injury; GSW, gunshot wound; MVC, motor vehicle collision; MCC, Motorcycle crash. * mean ± SD of age and median (interquartile-IQR) of days in the intervention.

**Table 2 ijerph-20-06958-t002:** List of antibodies used in fluorescent immunohistochemistry and immunoblotting.

Antibody	Type	Catalog Number	Vendor	Dilution	Host
MHC I (slow)	Primary antibody	NCL-MHCs	Leica	1:50	Rabbit
Alexa 594	Secondary antibody	A-11012	ThermoFisher Scientific	1:200	Goat
Anti-Laminin	Primary antibody	L8271-.2ML	Sigma-Aldrich	1:1000	mouse
Alexa 488	Secondary antibody	A-11001	ThermoFisher Scientific	1:200	Goat
MHC IIa (fast)	Primary antibody	NCL-MHCf	Leica	1:40	Rabbit
GLUT-4	Primary antibody	PA5-23052	Fisher Scientific	1:1000	Rabbit
AMPK-α	Primary antibody	2532S	Cell Signaling	1:1000	Rabbit
Phospho (Thr172)-AMPK-α	Primary antibody	2535S	Cell Signaling	1:1000	Rabbit
CaMKII	Primary antibody	4436S	Cell Signaling	1:1000	Rabbit
Phospho (Thr286)-CaMKII	Primary antibody	12716	Cell Signaling	1:1000	Rabbit
Akt	Primary antibody	9272S	Cell Signaling	1:1000	Rabbit
Phospho (Ser473)-Akt	Primary antibody	9271S	Cell Signaling	1:1000	Rabbit
AS160	Primary antibody	2447S	Cell Signaling	1:1000	Rabbit
Phospho (Thr642)-AS160	Primary antibody	4288S	Cell Signaling	1:1000	Rabbit
Hexokinase II	Primary antibody	2106S	Cell Signaling	1:1000	Rabbit
Glycogen Synthase (GS)	Primary antibody	3893S	Cell Signaling	1:1000	Rabbit
Phosphor (Ser641)-GS	Primary antibody	3891S	Cell Signaling	1:1000	Rabbit
IRS-1	Primary antibody	2390S	Cell Signaling	1:1000	Rabbit
Phospho (Tyr895)-IRS-1	Primary antibody	3070S	Cell Signaling	1:1000	Rabbit

## Data Availability

The data that support the findings of this study are available on request from the corresponding author, [CYF].

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
