# Peer review of "The Effect of Lower Limb Combined Neuromuscular Electrical Stimulation on Skeletal Muscle Signaling for Glucose Utilization, Myofiber Distribution, and Metabolic Function after Spinal Cord Injury"

_ijerph, 2023, doi:10.3390/ijerph20206958_

Round 1

Reviewer 1 Report (Previous Reviewer 1)

The authors presented a new improved version of the manuscript however there are important points to consider mainly in the methodology and results presentation.

Major comments: 

1. Myofiber-type Distribution. The authors describe an immunohistochemistry method to evaluate the myofiber-type distribution and use two studies (25 and 26) as references for the method.  

1.1. Reference number 25 does not correspond with an immunohistochemistry protocol.

1.2. The authors described the protocol used but there are lack of important details. Please consider adding a table with complete antibody (primary and secondary) information (catalog number, vendor, dilution, host). The authors could do the same for immunoblotting antibodies. The authors don`t explain how many sections, areas, or # of fibers were evaluated for each patient.

2. Immunoblotting

The authors wrote, "The density of the band was measured against the total lane background to obtain the band normalized volume". Why did the authors not use a reference protein (b actin, GAPDH, or other) to normalize the data? If not possible to use the reference protein method, could you add a reference to support your normalization method?

3. Statistical analysis

"Descriptive statistics for subjects’ characteristics were calculated with mean ± standard deviation for normally distributed continuous variables. Median and interquartile range (IQR) are reported for non-normally distributed variables". What normality tests (D`Agostino & Pearson test, Anderson-Darling test, Shapiro-Wilk test, Kolmogorov-Smirnov test, or other) do the authors apply to verify if the results present a normal or non-normal distribution before defining the specific statistics for each data?

4. The authors present the new version of the results using three groups: 'baseline, post-intervention, and within-group change'. What does the  "within-group change" mean? The authors could add a brief explanation from each group in the methods section. 

5. The authors mentioned using 15 samples for Skeletal Muscle Intracellular Signaling, however, in Figure 5, they showed Immunoblots for 3 controls and 5 Comb-NMES. Was the statistical analysis performed considering 8 or 15? If 15 samples were used, could you show as supplementary material all immunoblots used?

6. For Myofiber-type Distribution results, could you add a representative image from immunohistochemistry for each group to illustrate the graph?

Minor comments:

1. Standardize the writing of Type fibers using a single option Type  or type throughout the text.

2. Carefully review the lack of space (line 250, 382) 

3. Correction needed: fibres to fibers (line 386).

Author Response

Reviewer 2 Report (Previous Reviewer 3)

My comments have been addressed.

Few minor corrections/edits:

Not all of your x-axis on your figures match either in font, size, or potentially the angle (ie some look stretched out or at a slightly different angle) making them not consistent from graph to graph-seems to be mainly figure 4 and I cannot confirm if it is my pdf or not.

Line 389: I believe reducting should be reduction

Author Response

Response to Reviewer 2 Comments

My comments have been addressed.

Few minor corrections/edits:

Not all of your x-axis on your figures match either in font, size, or potentially the angle (ie some look stretched out or at a slightly different angle) making them not consistent from graph to graph-seems to be mainly figure 4 and I cannot confirm if it is my pdf or not.

Response: We have reformatted all the figures as suggested.

Line 389: I believe reducting should be reduction

Response: The word has been corrected.

Round 2

Reviewer 1 Report (Previous Reviewer 1)

The authors insert addition details in methods and results sections as suggested. Also, they include a representative fluorescent immunohistochemistry images from vastus lateralis (figure 7) to ilustrate the graph presented in figure 6.

The authors should be consider creat a single figure using the images (figure 7) and graph (figure 6). It is important insert visual information in the images indicating what they evaluate. (use arrows or other simbol) and add in the legend what is showed by red, blue. green colors. please, insert scale bar.

Author Response

The authors should be consider creat a single figure using the images (figure 7) and graph (figure 6). It is important insert visual information in the images indicating what they evaluate. (use arrows or other simbol) and add in the legend what is showed by red, blue. green colors. please, insert scale bar.

Response: Thank you for providing your valuable suggestion. The two figures have been combined into a single figure, and arrows have been incorporated into each fiber type inside the fluorescent images. The fibers were quantified both before and after the intervention in this study, applying the approaches described in lines 176–181

This manuscript is a resubmission of an earlier submission. The following is a list of the peer review reports and author responses from that submission.

Round 1

Reviewer 1 Report

Please see all comments in the attached file. The comments are in yellow highlight.

Reviewer 2 Report

 Alharbi et al. studied the effects of a combined neuromuscular electrical stimulation (Comb-NMES) regimen on skeletal muscle protein signaling for glucose uptake, myofiber type distribution, and metabolic function in individuals with spinal cord injury (SCI) within 14 days of the injury. The study aimed to determine whether early utilization of Comb-NMES could help maintain healthy myofiber type and metabolic function following SCI. The study is well-designed, methodologies are well-described, and the findings in muscle signaling and myofiber typing are interesting.

Several minor points:

1.      Figure 1. Need to explain the training process of the graph more clearly. The numbers of Repetition (green) and Weight (purple) in Figure 1A are difficult to read; the numbers of Duration (red) and Frequency (purple) in Figure 1B are difficult to read; a graph with a scale of lower number is required, or clear explanation is required.

2.      Figure 4. Representative immunoblots for the examined proteins should be provided.

3.      Figure 5. The type I and type IIx in baseline between two groups seem somewhat different. Are there any significant differences?

Reviewer 3 Report

Thank you for submitting your work to International Journal of Environmental Research and Public Health. Below I have provided a point by point review of each section followed by a general review of each section and/or the overall paper. My goal is to provide feedback that I believe will help improve the paper while maintaining the rigors of science and the Journal. I hope you find the review helpful, and thank you for your hard work.

Abstract:

Should state aerobic and resistant trained (Comb-NMES).

Introduction:

 Line 37: Missing citation

Line 70: Same as abstract, should mention aerobic and resistance trained

Materials and Methods:

Could you please provide the mean and SD for the days post injury that IPR and pre-discharge? As written it appears there could be up to a 4 week gap for some participants (i.e. 2 days post injury and 5 weeks pre-discharge). If this is not the case, please clarify.

Line 97: Please provide a statement about the distribution of days. You stated 3d/wk, but were these separated by a rest day or where they consecutive? Where they consecutive only when required based on other treatments etc?

Line 208: Can you please indicate which variables were log transformed? This will help with interpreting the figures

Results:

Line 236: I find this confusing, is Post-hoc here your post-hoc test, or are you simply stating this is being completed after your normal statistical tests? This should be clarified (also, see comment under figure)

Based on the results and the variability in the data, do your findings hold true if you express your data as a fraction of baseline?  This would provide a better control for the changes over time considering the heterogeneity of SCI in general (this is especially true for section 3.4).

Discussion:

 Line 358 is incredibly speculative. I understand the concept and why this is stated, but to make it a statement dependent on a comparative effectiveness study is an incredible stretch. Please consider revising.

Line 394: this seems in opposition of your hypothesis. How is this surprising if you state improvements in lipids are expected?

Section 4.3 and general: I find it surprising that nothing regarding the Henneman principle is addressed along with the fact that stimulating with an external source recruits all muscle fibers (See Ventilatory work from Dr. DiMarco).

Tables/Figures:

Table 1: Did you find differences based on incomplete v complete? And can you rule out natural recovery following injury. That is, did some individuals begin to regain additional motor function?

Why was SCI20 only in the intervention 9 days? You should consider computing an ANCOVA (or similar) to account for days in the intervention.

Figure 1: Can you explain the drop in duration for participant 33 (1b)? This does not seem to match your text.

Figure 2: I would suggest adding 2 more bars to this figure (and potentially all others) showing the change (i.e. -19.9 vs -7.3). As shown, I do not think your figures do your data justice. I would also consider indicating significant differences on your bar graphs since you’re still indicating there was a difference in the change. The addition of the 2 bars would resolve this issue.

Overall, I found the figures underwhelming for your findings. Please consider expressing as a fraction of baseline and/or adding a 3rd group of bar graphs for the change pre/post for each group. I would also consider labelling significant findings on these graphs based on your post-hoc/other tests so that the readers can immediately interpret your results from your graph(s)

Overall Impression:

 Overall, I find this as a very good manuscript with only minor concerns to be addressed. It is well written and with good detail. However, a few points of clarity are required.
